# An In Vitro Cell Culture Model for Pyoverdine-Mediated Virulence

**DOI:** 10.3390/pathogens10010009

**Published:** 2020-12-24

**Authors:** Donghoon Kang, Natalia V. Kirienko

**Affiliations:** Department of BioSciences, Rice University, Houston, TX 77005, USA; dk30@rice.edu

**Keywords:** *Pseudomonas aeruginosa*, pyoverdine, macrophages, virulence

## Abstract

*Pseudomonas aeruginosa* is a multidrug-resistant, opportunistic pathogen that utilizes a wide-range of virulence factors to cause acute, life-threatening infections in immunocompromised patients, especially those in intensive care units. It also causes debilitating chronic infections that shorten lives and worsen the quality of life for cystic fibrosis patients. One of the key virulence factors in *P. aeruginosa* is the siderophore pyoverdine, which provides the pathogen with iron during infection, regulates the production of secreted toxins, and disrupts host iron and mitochondrial homeostasis. These roles have been characterized in model organisms such as *Caenorhabditis elegans* and mice. However, an intermediary system, using cell culture to investigate the activity of this siderophore has been absent. In this report, we describe such a system, using murine macrophages treated with pyoverdine. We demonstrate that pyoverdine-rich filtrates from *P. aeruginosa* exhibit substantial cytotoxicity, and that the inhibition of pyoverdine production (genetic or chemical) is sufficient to mitigate virulence. Furthermore, consistent with previous observations made in *C. elegans*, pyoverdine translocates into cells and disrupts host mitochondrial homeostasis. Most importantly, we observe a strong correlation between pyoverdine production and virulence in *P. aeruginosa* clinical isolates, confirming pyoverdine’s value as a promising target for therapeutic intervention. This in vitro cell culture model will allow rapid validation of pyoverdine antivirulents in a simple but physiologically relevant manner.

## 1. Introduction

*Pseudomonas aeruginosa* is a Gram-negative, multidrug-resistant, nosocomial pathogen that frequently causes ventilator-associated pneumonia in intensive care units and chronic lung infections in cystic fibrosis patients [1,2]. This pathogen has a particularly diverse set of virulence determinants that facilitate host infection, including those that enable it to avoid immune recognition (e.g., elastases) [3,4] and obtain nutrients within the host (e.g., siderophores, proteases, lipases, etc.) [5]. It also possesses acute virulence factors that damage host cells or tissues, including the type III secretion system [6], and chronic virulence factors that support pathogen colonization, such as quorum-sensing and biofilm formation [7]. However, certain virulence factors, such as the siderophore pyoverdine, have multifaceted roles in *P. aeruginosa* pathogenesis, making them promising targets for therapeutic intervention.

As a siderophore, pyoverdine exhibits exceptionally high affinity towards ferric iron and is able to remove the metal from host ferroproteins such as transferrin and lactoferrin [8]. Under physiological conditions, where the host actively restricts iron availability, pyoverdine production is necessary for *P. aeruginosa* iron uptake and growth [9,10,11]. Pyoverdine production also promotes *P. aeruginosa* biofilm formation [12], which is consistent with the established paradigm that iron availability is necessary for biofilm development [12,13].

Pyoverdine also regulates the expression of secreted toxins in *P. aeruginosa*. Recognition of ferripyoverdine by its outer membrane receptor, FpvA, releases the sequestration of the alternate sigma factor PvdS, allowing it to upregulate genes responsible for the production of the translational inhibitor exotoxin A, the protease PrpL, and pyoverdine itself [14,15,16]. Exotoxin A in particular has been recognized as one of the most potent toxins secreted by *P. aeruginosa*, as it can induce apoptotic death in host cells and kill model organisms [17,18,19]. PrpL (also known as protease IV) is an extracellular protease that degrades host factors necessary for pulmonary mucosal immunity, such as surfactant proteins and interleukin-22 [20,21,22].

We have demonstrated that pyoverdine exerts virulence against the model nematode *Caenorhabditis elegans* even in the absence of the pathogen [23]. Pyoverdine translocated from the intestinal lumen into host tissue, disrupting iron and mitochondrial homeostasis [24,25]. Pyoverdine also inhibited redox metabolism and ATP synthesis, and activated mitochondrial quality control pathways [24,26,27]. Some combination of these functions makes pyoverdine critical for *P. aeruginosa* virulence in invertebrate hosts (such as *C. elegans* and *Galleria mellonella*) [23,28] and in murine infection [5,10,29].

Several antivirulence strategies have been developed to mitigate pyoverdine-mediated virulence. Notably, 5-fluorocytosine (5-FC), an inhibitor of pyoverdine biosynthesis, attenuates *P. aeruginosa* pathogenesis in *C. elegans* and mice [30,31,32,33]. Other fluoropyrimidines, including 5-fluorouridine (5-FUR) and 5-fluorouracil (5-FU), also have this ability; likely all of them function through the common metabolic intermediate 5-FUR [30]. Other agents that have proven to limit pyoverdine production include twin-arginine translocase inhibitors [34,35,36], quorum-sensing inhibitors [37,38,39], and biofilm inhibitors [40]. Pyoverdine-mediated virulence can also be attenuated by gallium (III) nitrate. Gallium has a nearly identical ionic radius to iron and is bound by pyoverdine almost as strongly as iron (III) [41]. However, because gallium (III) is redox inactive, it cannot be removed from the siderophore, causing it to function as a suicide inhibitor. For this reason, gallium (III) rescues *G. mellonella* or *C. elegans* from *P. aeruginosa* pathogenesis [31,42]. We also recently reported several molecules that block pyoverdine function by interacting with the chromophore region of the siderophore [43,44]. These inhibitors decreased the expression of PvdS-regulated genes and rescued *C. elegans* from *P. aeruginosa* pathogenesis. However, most of these treatments have yet to be validated in mammalian models, partly due to a relative paucity of accessible systems for testing.

In this report, we introduce a robust, in vitro cell culture model where murine macrophage cytotoxicity is associated with exposure to pyoverdine. We demonstrate that spent growth media, which is rich in pyoverdine, exhibits substantial toxicity against cells. Genetic or chemical inhibition of pyoverdine production is sufficient to mitigate this effect. We also demonstrate that pyoverdine translocates into macrophages and disrupts mitochondrial homeostasis, consistent with our previous observations in *C. elegans*. Importantly, pyoverdine-mediated virulence was observed not only for laboratory-adapted strains of *P. aeruginosa*, but also in clinical isolates from cystic fibrosis patients, where pyoverdine production strongly correlated with cytotoxicity. This model allows rapid virulence assessment and is amenable to high-throughput applications. In addition, this approach will facilitate investigation of the molecular functions of pyoverdine at the mechanistic level in a more biomedically-relevant context.

## 2. Results

### 2.1. Pyoverdine Production Is Important for P. aeruginosa Virulence against Murine Macrophages

To investigate whether pyoverdine production contributes to pathogen virulence in an in vitro cell culture model, we grew *P. aeruginosa* PAO1 in serum-free cell culture media (Eagle’s Minimal Essential Medium, EMEM). As anticipated from the low iron content of this media, *P. aeruginosa* produced substantial amounts of pyoverdine (Figure 1A,B). The levels were comparable to those seen from *P. aeruginosa* grown in low-iron bacterial media conditions that are commonly used for pyoverdine production (Appendix A). Importantly, an isogenic mutant deficient in pyoverdine production (PAO1*ΔpvdF*) grew to a similar optical density in EMEM, despite producing virtually no detectable pyoverdine.

RAW264.7 murine macrophages were exposed to spent, cell-free media (hereafter referred to as “filtrate”, see Section 4 for details) from wild-type and pyoverdine-deficient *P. aeruginosa* cultures to examine these materials’ ability to kill mammalian cells. Macrophages treated with wild-type filtrate showed substantial cell death, as measured using the alamarBlue cell viability assay, while cells treated with pyoverdine-deficient filtrate had significantly better survival (Figure 1C). We validated these results using another cell viability assay that uses a combination of cell-permeable (Hoechst 33342) and cell-impermeable (Sytox Orange) dyes to identify living and dead cells (Figure 1D), indicating that pyoverdine production is associated with cytotoxicity in macrophages. We observed similar results in two other murine macrophage cell lines, J774A.1 and the alveolar macrophage cell line MH-S (Appendix A).

Importantly, filtrate toxicity was independent of pyochelin, another major siderophore produced by *P. aeruginosa*. The pyochelin biosynthetic mutant (PAO1*ΔpchBA*) did not exhibit significant differences in bacterial growth, pyoverdine production, or filtrate toxicity from that of wild-type bacteria (Appendix A). Interestingly, disruption of pyochelin production in the pyoverdine mutant was sufficient to deprive the pathogen of iron, inhibiting bacterial growth, and attenuating virulence. Furthermore, although pyocyanin, a toxin known to induce oxidative stress in host cells [45,46], was present in filtrates at concentrations of 4.3 ± 0.6 µM, it was insufficient to induce macrophage death within the timeframe of pyoverdine-mediated damage, as demonstrated with purified pyochelin (Appendix A).

We further validated our in vitro pathogenesis model by treating macrophages with bacterial filtrates from *P. aeruginosa* grown in the presence of the pyoverdine biosynthetic inhibitors 5-fluorocytosine (5-FC) or 5-fluorouridine (5-FUR), which were added prior to inoculation [30,32]. Under our growth conditions, these drugs effectively inhibited pyoverdine production without overtly affecting bacterial growth (Appendix A). Inhibition of pyoverdine production by these compounds attenuated filtrate toxicity for wild-type *P. aeruginosa* (Figure 1E). Fluoropyrimidines had no apparent effect on cell viability for macrophages treated with filtrate from the biosynthetic mutant. This is consistent with our previous observations that the mitigation of virulence by these fluoropyrimidines functions solely by limiting pyoverdine production [30].

In a previous study, we demonstrated that pyoverdine directly exerts virulence against *C. elegans* by treating hosts with bacterial filtrates that had been autoclaved to inactivate heat-labile materials, which includes most proteins. However, this treatment could have damaged other molecules as well. To rule out the possibility that other molecules were responsible for the cytotoxicity seen in macrophages, filtrates were passed through a 5 kDa centrifugal filter to remove large macromolecules. Pyoverdine produced by *P. aeruginosa* PAO1, often referred to as Type I pyoverdine, has a molecular weight of approximately 1,350 Da. As expected, pyoverdine passed through the membrane with relative ease; ~75% of the initial pyoverdine was found in this low-molecular-weight fraction (hereafter referred to as “flowthrough”) as judged by fluorescence (Appendix A). Cell viability was measured in macrophages that were exposed to this low-molecular-weight material from wild-type *P. aeruginosa* or the pyoverdine biosynthetic mutant. As expected, the flowthrough from wild-type bacteria had greater cytotoxicity, although removing large macromolecules reduced cell death for both genotypes (Figure 1F). We hypothesized that the difference in the ability of flowthrough from wild-type and pyoverdine mutant to kill macrophages is likely due to direct cytotoxicity from pyoverdine (i.e., it is independent of proteinaceous virulence factors upregulated by pyoverdine). Consistent with this hypothesis, pre-incubation of the flowthrough with equimolar concentrations of gallium (III) nitrate reduced the cytotoxicity of wild-type material by 40% (Figure 1F). However, even in the presence of gallium, there was considerable difference between the cells treated with wild-type and pyoverdine mutant flowthrough, suggesting that pyoverdine is not the only factor contributing to cell death.

### 2.2. Pyoverdine Translocates into Macrophages and Disrupts Mitochondrial Function

The likeliest explanation for the cell death we observed is that pyoverdine is interacting with the cellular iron pool. Depletion of iron from the medium is unlikely to cause these levels of cell death in the time frame examined. In order to have this effect, though, pyoverdine would first need to gain entry into cells. To determine whether pyoverdine was within cells, macrophages were treated with flowthrough for 2 h and then washed to remove extracellular pyoverdine. Cells were then lysed in the presence of excess 8-hydroxyquinoline (8HQ). Excess 8HQ strips iron from pyoverdine, restoring pyoverdine fluorescence that had been quenched by iron-binding before or during cell lysis [47]. We observed substantial pyoverdine fluorescence in macrophage cell lysates treated with the flowthrough from wild-type *P. aeruginosa* but not in cells treated with material from the pyoverdine mutant (Figure 2A). This fluorescence was quenched by the addition of ferric iron, indicating that it originated from pyoverdine.

We also visualized pyoverdine’s innate fluorescence within cells using confocal microscopy. Macrophages exposed to flowthrough from wild-type bacteria accumulated substantial intracellular pyoverdine (Figure 2B, upper left). In contrast, cells treated with material from pyoverdine-deficient mutants (Figure 2B, upper right, Figure 2C) or filtrate quenched with ferric iron (Figure 2B, lower left, Figure 2C) showed no fluorescence in the channel characteristic for pyoverdine. Chelating gallium (III) is known to considerably increase the quantum yield of pyoverdine [48]. As anticipated, pre-incubating pyoverdine with gallium (III) increased the fluorescence yield of cells treated with wild-type, but not pyoverdine-deficient material (Figure 2B, lower right, Figure 2C, Appendix A). This difference in fluorescence between samples treated with pyoverdine and pyoverdine with gallium may also be partly explained by the quenching of pyoverdine fluorescence upon the chelation of intracellular iron. In either case, gallium chelation did not diminish the ability of the siderophore to translocate into mammalian cells.

To rule out the possibility that the fluorescence observed was due to accumulation of pyoverdine on the cell surface, we counterstained the plasma membrane and nucleus in macrophages treated with the pyoverdine–gallium complex. The increased fluorescence yield allowed us to visualize pyoverdine, despite its partial overlap in emission spectra with that of the nucleic acid counterstain. Pyoverdine fluorescence was also distinct from that of the plasma membrane (Figure 2D), indicating that pyoverdine is localized to a subcellular compartment, perhaps a sorting compartment like the early endosome.

We tested this localization by exposing macrophages to both pyoverdine and Texas Red-labeled dextran (10,000 MW), which macrophages internalize via macropinocytosis. Dextran of this molecular weight range is well-established as a tool for visualizing early endosomes [49]. We observed extensive colocalization of pyoverdine and dextran (Figure 2E), indicating that most of the pyoverdine was likely in this cellular compartment. However, we did see some areas that only had pyoverdine fluorescence, which may indicate that the siderophore is also present elsewhere in the cell.

Based on its translocation into cells and its exceptionally high affinity for iron (higher than most host ferroproteins), we hypothesized that pyoverdine was likely to be hijacking host iron. One expected consequence of this activity is the disruption of mitochondrial function such as the fragmentation of mitochondrial networks which can be visualized using MitoTracker Green FM dye. For instance, cells treated with the mitochondrial uncoupler carbonyl cyanide 3-chlorophenylhydrazone (CCCP) exhibit such fragmentation (Figure 3A). We treated macrophages with flowthrough from wild-type PAO1 or the *ΔpvdF* mutant and evaluated the effects of pyoverdine on host mitochondrial networks. Mitochondrial networks in macrophages treated with wild-type flowthrough exhibited substantially greater fragmentation compared to those in cells treated with material from the pyoverdine biosynthetic mutant (Figure 3B). These findings are consistent with our previous results where pyoverdine disrupts *C. elegans* mitochondrial homeostasis [24,25].

### 2.3. Pyoverdine Production Correlates to Virulence in P. aeruginosa Clinical Isolates

Finally, we investigated the role of pyoverdine production in the virulence of *P. aeruginosa* isolates obtained from patients with cystic fibrosis. Previously, we compared the virulence of strains that exhibited high (PA2-61, PA2-72) or low (PA2-88, PA3-22) levels of pyoverdine production in *C. elegans* [31]. For comparison, we tested the amount of pyoverdine produced by these strains in Liquid Killing (LK) media and in serum-free EMEM (Figure 4A,B). Exposure to the high-pyoverdine producing strains PA2-61 or PA2-72 was much more lethal to *C. elegans* in the Liquid Killing assay than to their low-pyoverdine producing counterparts, PA2-88 or PA3-22 (Figure 4C). A similar pattern was seen in macrophages. Bacterial filtrates from PA2-61 and PA2-72 were more toxic to macrophages than identically-prepared materials from PA2-88 and PA3-22, which caused virtually no cytotoxicity (Figure 4D). As we observed for PAO1, 5-FC inhibited pyoverdine production in the two virulent isolates, attenuating filtrate cytotoxicity (Figure 4D).

To more broadly explore the relationship between pyoverdine and virulence, we expanded our survey to include 19 additional *P. aeruginosa* isolates from various sources, including burn wounds, urinary tract infections, and plants [50]. These isolates displayed heterogeneity with regard to pyoverdine production, though most strains exhibited significant levels of pyoverdine (at least 30% of that of PAO1, Figure 5A). As expected, we observed a strong correlation between pyoverdine levels and filtrate cytotoxicity (Figure 5B). Importantly, this correlation was not due to differences in bacterial growth (Figure 5C,D). These results reaffirm the role of pyoverdine in our in vitro macrophage pathogenesis model and indicate that pyoverdine may be a useful target for clinical intervention.

## 3. Discussion

In response to the rising prevalence of bacterial drug resistance, increased interest has been shown in the investigation of alternative treatment approaches, including inhibiting bacterial virulence. Compounds with this function generally prevent either the biosynthesis, the secretion, or the function of their target. Importantly, antivirulents are likely to fall into two different categories: pure antivirulents, which do not overtly compromise pathogen growth or survival, and mixed-function compounds, which may inhibit both virulence and bacterial growth by either the same or different mechanisms. It is postulated that the reduced effect on bacterial growth by pure antivirulents might limit the selective pressure on pathogens to develop resistance, which could extend the long-term utility of these drugs.

Model organisms, including *C. elegans*, mammalian cells, and mice have facilitated the development of *P. aeruginosa* antivirulents. Work in mice and *C. elegans* has established an important role for pyoverdine in virulence [5,23,31,51], and validated antivirulents that were identified in various biochemical screens. For instance, Imperi and colleagues identified 5-fluorocytosine from a screen for pyoverdine biosynthetic inhibitors and validated its therapeutic properties in a murine lung infection model [32]. Our lab has identified several compounds that inhibit pyoverdine function and rescue *C. elegans* hosts during *P. aeruginosa* pathogenesis [43]. Inhibitors of the type III secretion system, elastase LasB, and quorum-sensing have also been identified and validated using these model pathosystems [52,53,54,55].

Due to their inexpensive maintenance conditions, rapid generation time, and small size, *C. elegans* has been a robust tool for the development of high-content, high-throughput screens for host–pathogen interactions. For instance, we performed a screen to identify molecules that rescue *C. elegans* during *P. aeruginosa* Liquid Killing and identified pyoverdine biosynthetic inhibitors such as fluoropyrimidines and inhibitors of pyoverdine function. *C. elegans*-based platforms have also been used to discover novel anti-infectives against *Enterococcus faecalis*, *Staphylococcus aureus*, and *Candida albicans* [56,57,58]. However, while *C. elegans* are amenable to high-throughput screens, they will not completely recapitulate host–pathogen interactions seen in mammalian hosts. Notably, *C. elegans* are immune to *P. aeruginosa* type III secretion [59], one of the most important acute virulence factors in the pathogen [6].

While validation of anti-infectives using murine-infection models is the best approach, it is also expensive and comes with an ethical burden. It may also be impractical to test a large number of therapeutic leads. Establishing intermediary pathogenesis assays, such as monolayer or three-dimensional cell culture models [60,61] may prove invaluable for quickly evaluating the therapeutic potential of novel drugs.

Thus far, several in vitro models have been developed for type III secretion-mediated virulence in *P. aeruginosa*. For instance, early work has established that effectors injected by the type III secretion system, such as ExoT or ExoU, induce apoptotic or necrotic death in mammalian cells, respectively [62,63]. In *P. aeruginosa* macrophage infection models, the pathogen activates the NLRC4/IPAF inflammasome via flagellin and the type III secretion system to induce pyroptosis [64,65,66]. *P. aeruginosa* can also activate the NLRP3 inflammasome to mitigate phagocytic killing by macrophages [67]. In endothelial cell infection models, the type III secretion system and the elastase LasB disrupt endothelial cell junction permeability, a key mechanism of bacterial dissemination within the host [68,69]. Another potentially useful set of in vivo models for researching *P. aeruginosa* virulence utilizes amoebal species such as *Acanthamoeba castellanii* and *Dictyostelium discoideum* [70,71].

To the best of our knowledge, we report here the first model for pyoverdine-mediated virulence in mammalian cell culture. Pyoverdine has been well-established as a key virulence factor in *P. aeruginosa*. Pyoverdine’s ability to provide iron for the pathogen and regulate downstream effectors such as exotoxin A and PrpL has been attributed to bacterial virulence in the mouse lung [5]. While our model differs from more conventional *P. aeruginosa* infection assays by focusing solely on secreted bacterial factors, we have recapitulated results obtained from *C. elegans* Liquid Killing and murine lung infection models. Moreover, this in vitro cell culture model provided unique insights into the effects of pyoverdine intoxication. While we previously demonstrated that pyoverdine translocates from the *C. elegans* intestinal lumen into neighboring tissue, we were unable to definitively prove that pyoverdine was present inside host cells [24]. In contrast, demonstrating internalization in murine macrophages was much simpler, and we were able to determine that pyoverdine is likely to localize to early endosomal compartments and visualize the mitochondrial damage that results from exposure. Future work in this model, and the wealth of cell biological tools available, may help elucidate the interactions between pyoverdine and intracellular iron sources, including mitochondrial ferroproteins. We are currently investigating whether other cell lines, including lung epithelial cells, also show pyoverdine-dependent cytotoxicity to more easily understand the effects of pyoverdine in the context of cystic fibrosis or ventilator-associated pneumonia.

## 4. Materials and Methods

### 4.1. Bacterial Strains and Growth Conditions

*P. aeruginosa* PAO1 and pyoverdine biosynthetic mutant PAO1*ΔpvdF* were provided by Dr. Dieter Haas. *P. aeruginosa* cystic fibrosis isolates PA2-88, PA3-22, PA2-61, and PA2-72 were provided by Dr. Carolyn Cannon [31]. The remaining clinical and environmental isolates in Figure 5 were provided by Dr. Frederick Ausubel [50].

To generate pyoverdine-rich, spent bacterial cultures, called “filtrates” herein, *P. aeruginosa* overnight culture grown in LB medium was diluted 20-fold into 2 mL of Eagle’s Minimum Essential Medium (EMEM) (Millipore Sigma, Burlington, MA, USA) in 6-well plates. The plate was covered with an air-permeable membrane and bacteria were grown statically at 37 °C for 16 h. Bacteria were then pelleted, and the supernatant was filtered through a 0.22 µm membrane. The filtrate was treated with 150 µg/mL amikacin to kill any remaining bacteria. To remove large biomolecules, this material was then filtered through a Vivaspin 5000 MWCO PES membrane centrifugal concentrator (Sartorius, Gottingen, Germany) at 3600 RCF, yielding the material herein referred to as “flowthrough”.

To chemically inhibit pyoverdine production, EMEM was supplemented with 50 µM 5-fluorocytosine (5-FC) or 10 µM 5-fluorouridine (5-FUR) prior to bacterial inoculation.

### 4.2. Macrophage Cell Lines and Growth Conditions

RAW264.7 and J774A.1 murine macrophages and MH-S murine alveolar macrophages were purchased from ATCC (Manassas, VA, USA). All cell lines were grown and passaged in RPMI medium supplemented with 10% bovine calf serum and penicillin/streptomycin (Millipore Sigma, Burlington, MA, USA) at 37 °C and 5% CO_2_ in a Symphony air-jacketed incubator (VWR, Radnor, PA, USA).

### 4.3. Cell Viability Assay

To perform the alamarBlue cell viability assay, approximately 2 × 10^6^ cells were seeded into each well of a 12-well plate (1 mL/well) and grown to > 95% confluency. The growth medium was then aspirated and replaced with bacterial filtrate. After filtrate exposure, 100 µL of alamarBlue HS Cell Viability Reagent (Invitrogen, Carlsbad, CA, USA) was added to each well and cells were incubated for an additional 1 h. The supernatant was then collected and briefly centrifuged to remove cells. Then, 100 µL of supernatant was transferred to a 96-well plate and resorufin fluorescence (Ex. 560 nm; Em. 590 nm) was measured using a Cytation5 multimode reader (Biotek, Winooski, VT, USA). Cell viability was calculated as a percentage of fluorescence measured from cells treated with bacterial filtrates compared to media control.

For Hoechst 33342 and Sytox Orange staining, approximately 2 × 10^5^ cells were seeded and adhered onto each well of a 96-well black clear bottom plate (100 µL/well). Cells were then stained with 8 µM Hoechst 33342 (Thermo Fisher Scientific, Waltham, MA, USA) in RPMI medium for 30 min. Afterwards, the Hoechst solution was aspirated and replaced with bacterial filtrate supplemented with 2.5 µM Sytox Orange (Invitrogen, Carlsbad, CA, USA). After filtrate exposure, cells were gently washed in RPMI medium and imaged using a Cytation5 multimode reader (Biotek, Winooski, VT, USA).

### 4.4. Measuring Pyoverdine Content

To measure pyoverdine production by *P. aeruginosa*, 100 µL of bacterial culture was transferred to a 96-well plate, and pyoverdine fluorescence (Ex. 405 nm; Em. 460 nm) was measured using a Cytation5 multimode reader (Biotek, Winooski, VT, USA).

To measure pyoverdine content in macrophages, cells were grown in T25 flasks to > 95% confluency. The growth medium was then aspirated and replaced with low-molecular-weight material from bacterial filtrates. After 2 h treatment, cells were scraped off, washed in phosphate-buffered saline (PBS) to remove exogenous pyoverdine, and resuspended in PBS. The cell suspension was mixed with 2 M 8-hydroxyquinoline in chloroform at a 1:1 (v/v) ratio, vortexed for 1 min, and incubated at room temperature for 8 h on a tube rotator. The mixture was then centrifuged at 16,000 RCF for 5 min. Pyoverdine fluorescence in the aqueous layer was measured using a Cytation5 multimode reader (Biotek, Winooski, VT, USA).

### 4.5. Confocal Microscopy

Approximately 1.25 × 10^6^ cells were seeded and adhered onto each well of a Lab-Tek II 8-well chambered cover glass (Thermo Fisher Scientific, Waltham, MA, USA) (500 µL/well). To visualize pyoverdine fluorescence within macrophages, cells were treated with flowthrough material from bacterial filtrates supplemented with 250 µM FeCl_3_, Ga(NO_3_)_3_, or solvent control and washed in RPMI medium to remove excess material. Pyoverdine fluorescence was visualized via a 405 nm laser-line using channel conditions for Pacific Blue on a Zeiss LSM 800 confocal laser-scanning microscope (Zeiss, Oberkochen, Germany). All micrographs were processed using Airyscan 2D SR processing to acquire super-resolution images. Pyoverdine fluorescence in each cell was quantified by its mean fluorescence intensity using ZEN image analysis software.

To label the cell nucleus and plasma membrane, cells were treated with 5 µM Syto 9 green fluorescent nucleic acid stain (Invitrogen, Carlsbad, CA, USA) and 5 µg/mL CellMask deep red plasma membrane stain (Invitrogen, Carlsbad, CA, USA) in RPMI medium for 30 min after pyoverdine exposure. Syto 9 fluorescence was visualized via a 488 nm laser-line using channel conditions for FITC. CellMask fluorescence was visualized via a 640 nm laser-line using channel conditions for Alexa Fluor 660. To label endocytic compartments, low-molecular-weight filtrate was supplemented with 250 µg/mL dextran Texas Red, 10,000 MW (Invitrogen, Carlsbad, CA, USA). Dextran fluorescence was visualized via a 561 nm laser-line using channel conditions for Texas Red. All micrographs were processed using Airyscan 2D SR processing to acquire super-resolution images.

To visualize mitochondrial networks, cells were first treated with flowthrough then stained with 500 nM MitoTracker Green FM (Invitrogen, Carlsbad, CA, USA) in RPMI medium for 30 min. Cells were washed in RPMI medium to remove excess dye. MitoTracker Green fluorescence was visualized via a 488 nm laser-line using the channel conditions for eGFP. All micrographs were processed using Airyscan 2D SR processing to acquire super-resolution images.

### 4.6. C. elegans Liquid Killing

*C. elegans* Liquid Killing (LK) was performed as previously described [72,73]. In brief, worms were treated with *P. aeruginosa* in LK media (25% SK media (0.3% NaCl, 0.35% Bacto Peptone in water) in S Basal buffer (100 mM NaCl, 50 mM potassium phosphate, pH 6.0)) in 384-well plates. After pathogen exposure, worms were stained with Sytox Orange (Invitrogen, Carlsbad, CA, USA) to label dead organisms. Fluorescent images were acquired using a Cytation5 multimode reader (Biotek, Winooski, VT, USA) and analyzed using CellProfiler.

## Figures and Tables

**Figure 1 pathogens-10-00009-f001:**
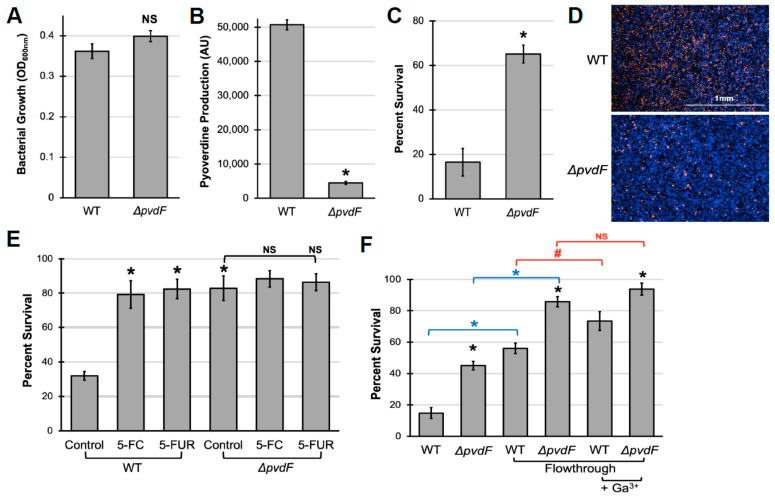
Pyoverdine production is important for virulence in murine macrophages. (**A**,**B**) Bacterial growth (**A**) and pyoverdine production (**B**) of wild-type *P. aeruginosa* PAO1 or the pyoverdine biosynthetic mutant PAO1*ΔpvdF* after 16 h incubation in Eagle’s Minimal Essential Medium. (**C**) RAW264.7 murine macrophage viability after 1.5 h exposure to filtrates from wild-type PAO1 or PAO1*ΔpvdF*. Cell viability was measured using an alamarBlue assay. (**D**) Visualization of macrophage cell death following filtrate exposure using counterstaining with Sytox Orange, a cell-impermeant nucleic acid stain (red). Cells were prelabeled with Hoechst 33342 (blue) then treated with filtrate in the presence of Sytox Orange for 1 h. (**E**) Macrophage viability after 1.5 h exposure to filtrates from bacteria grown in the presence of 50 µM 5-fluorocytosine (5-FC) or 10 µM 5-fluorouridine (5-FUR). (**F**) Macrophage viability after 2.5 h exposure to bacterial filtrates from wild-type PAO1 or PAO1*ΔpvdF* before and after the removal of large biomolecules by centrifugal filtration (<5 kDa flowthrough) or after pre-saturating the pyoverdine in the flowthrough with 250 µM gallium (III) nitrate. Error bars represent SEM from at least three biological replicates. * corresponds to *p* < 0.01, # corresponds to *p* < 0.05, and NS corresponds to *p* > 0.05 based on Student’s *t*-test.

**Figure 2 pathogens-10-00009-f002:**
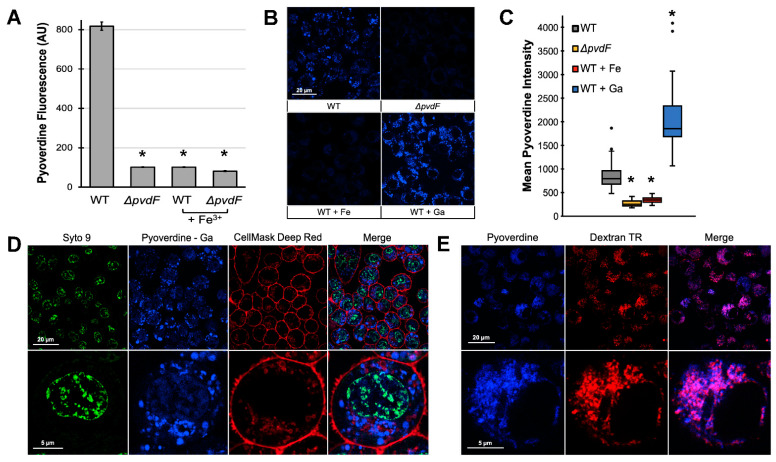
Pyoverdine translocates into macrophages. (**A**) Pyoverdine fluorescence in macrophage cell lysates after 2 h exposure to low-molecular weight flowthrough from bacterial filtrates. Lysate fluorescence was quenched by exogenous ferric iron. (**B**) Pyoverdine fluorescence within macrophages visualized via confocal laser-scanning microscopy after 1.5 h exposure to flowthrough pretreated with iron (III) chloride, gallium (III) nitrate, or solvent control. (**C**) Quantification of pyoverdine fluorescence within 50 individual cells. (**D**) Pyoverdine–gallium fluorescence within cells labeled with Syto 9 green fluorescent nucleic acid stain and CellMask deep red plasma membrane stain. The bottom row shows an enlarged micrograph of one representative cell. (**E**) Pyoverdine and dextran fluorescence within macrophages were visualized after 1.5 h exposure to wild-type flowthrough supplemented with dextran-Texas Red (10,000 MW)**.** The bottom row shows an enlarged micrograph of one representative cell. Error bars in (**A**) represent SEM from three biological replicates. * corresponds to *p* < 0.01 based on Student’s *t*-test. Cells labeled with Syto 9, CellMask, or dextran in the absence of pyoverdine are shown in Appendix A.

**Figure 3 pathogens-10-00009-f003:**
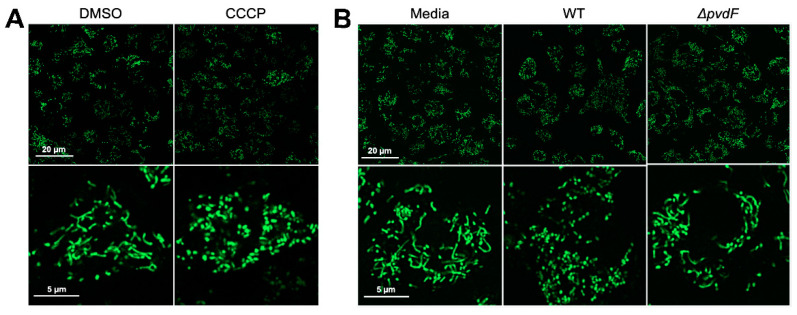
Pyoverdine production disrupts host mitochondrial networks. (**A**) Visualization of mitochondrial morphology in murine macrophages via MitoTracker Green FM staining after 2.5 h exposure to 50 µM CCCP or DMSO solvent control. (**B**) MitoTracker Green FM staining in macrophages after 1 h exposure to flowthrough. Bottom row shows enlarged micrograph of a representative cell.

**Figure 4 pathogens-10-00009-f004:**
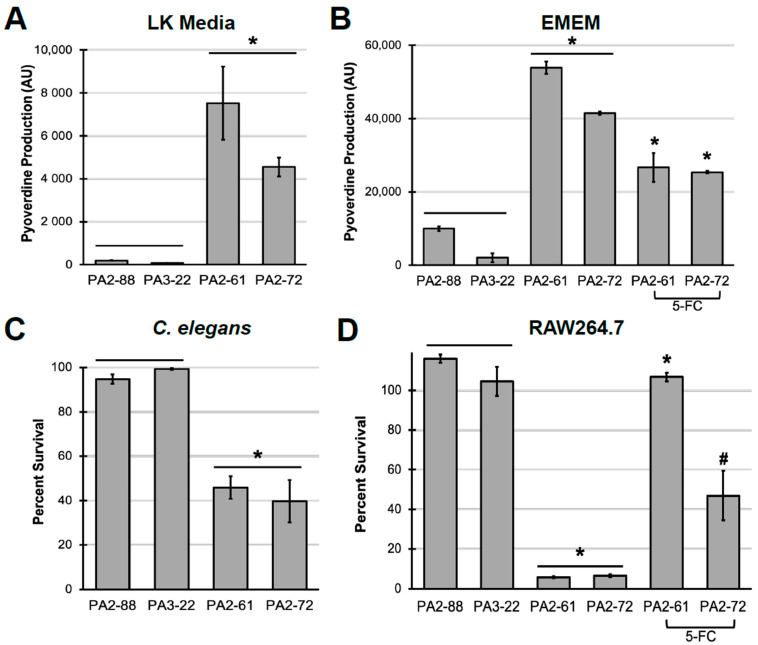
In vitro macrophage pathogenesis model recapitulates results from *C. elegans*. (**A**) Pyoverdine production by *P. aeruginosa* cystic fibrosis isolates after growth in Liquid Killing media in the presence of *C. elegans.* (**B**) Pyoverdine production by *P. aeruginosa* cystic fibrosis isolates in Eagle’s Minimum Essential Medium with or without 50 µM 5-fluorocytosine (5-FC). (**C**) *C. elegans* survival after exposure to *P. aeruginosa* under Liquid Killing conditions. (**D**) Macrophage survival after exposure to bacterial filtrates from *P. aeruginosa*. Error bars represent SEM from three biological replicates. * corresponds to *p* < 0.01 and # corresponds to *p* < 0.05 based on Student’s *t*-test.

**Figure 5 pathogens-10-00009-f005:**
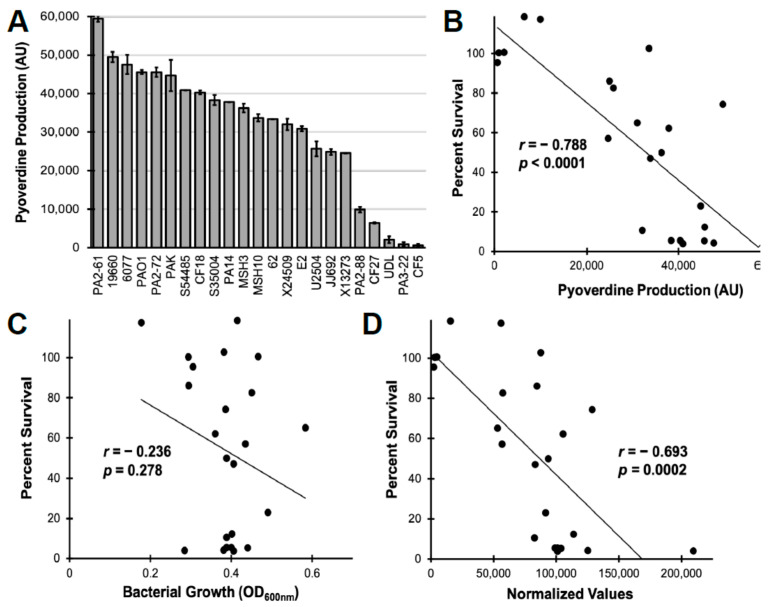
Pyoverdine production correlates with pathogen virulence in *P. aeruginosa* clinical and environmental isolates. (**A**) Pyoverdine production by a panel of 23 *P. aerugionosa* isolates after 16 h incubation in serum-free Eagle’s Minimum Essential Medium. (**B**–**D**) Correlation between pyoverdine production (**B**), bacterial growth (**C**), or pyoverdine normalized to growth (**D**) and filtrate toxicity against murine macrophages. Error bars represent SEM from two biological replicates. Each point represents the average of two biological replicates.

## Data Availability

Data available upon request.

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
