# Peer review of "An In Vitro Cell Culture Model for Pyoverdine-Mediated Virulence"

_pathogens, 2020, doi:10.3390/pathogens10010009_

Round 1
Reviewer 1 Report
The study by Kang et.al. presents the role of pyoverdine in the virulence of P. aeruginosa, a human pathogen of clinical significance. The authors established the cytotoxicity of pyoverdine by using bacterial supernatants and several lines of murine macrophages. They also monitored pyoverdine uptake and potential association with endosomes and mitochondrial integrity. Finally, they correlated the levels of pyoverdine production and cytotoxicity in a panel of 19 isolates.
The manuscript is well written and provides useful advances in using pyoverdine as a measure for P.a. virulence and cytotoxicity against murine macrophages. Validating the impact of pyoverdine on macrophages by alternative assays and three different cell lines is appreciated.
Below are several additional questions and suggestions.
1.The need for the intermediary model system is well explained in the discussion but could be better delivered in the abstract.
- The observations in Fig. 1D need to be explained in the text.
3.L. 94-100 Following the results here raises the question of whether the level of pyoverdine was tested in the filtrates. You may want to consider moving these data up here. Do you think pyocyanin could also contribute to the cytotoxicity?
- Pyoverdine and growth data in most of the figures are shown at one time point, without clarifying what that time point that is. This information needs to be provided at least in the methods and the corresponding figure legends. Not knowing the timing of cell harvest makes it difficult to compare different strains, as their growth phases may be different, which would impact pyoverdine production.
- Please describe how the cultures were treated with the inhibitors (Fig 1E)
- L.142 I would not call 1.6 fold “almost half” and just provide an accurate comparison.
- L. 150 It says 2 h here, but 1.5 h in the legends.
- L.157 Were the cells washed prior to microcopy to remove potentially attached pyoverdine?
- L 162-168 This part on gallium is confusing. What is meant by “quantum yield”, production or availability? Is gallium present in any quantitates in the medium, so it needs to be chelated? Why would the presence of gallium increase the fluorescence of pyoverdine? In my opinion, these data with gallium do not add any insight to this paper.
- L.171 Why were the cells treated with gallium here? Does it have anything to do with the pyoverdine translocation across the membranes? Is the following experiment (L.174) with dextran also involves treatment with gallium?
- L.197 Should it be wild-type PAO1 and pvdF mutant?
- For mitochondrial fragmentation, do you have any positive control with some treatment known to increase such fragmentation? Is there any way to quantify the fragmentation using the collected micrographs?
- L. 212-218 Spell out LK and include into the methods. The killing assay needs to be described in the method section. Was it done with bacterial cells or their supernatants? Fig. 4D How could survival % be greater than 100? How would you explain the difference in the profiles of 5-FC-treated cells in 4D vs their pyoverdine profile in 4B?
- L. 227-235 Did you see any correlation between the production of pyoverdine and the source of isolation? How was the bacterial growth assessed here: at which time point was the OD measured? Did these measurements represent the same growth phase in each isolate? Do you think it would be better to normalize pyoverdine production by OD first and then use the normalized data for the correlation in 5B?
- Methods. L. 312 Why was it needed to kill the remaining P.a. if the following step was filtering? Section 4.4 needs to be extended to describe measuring pyoverdine in bacterial cultures and filtrates. As above mentioned, the time points need to be provided.
Reviewer 2 Report
The authors of this study clearly demonstrate that pyoverdine, the high-affinity siderophore of Pseudomonas aeruginosa enters murine macrophages and cause cell death by disrupting the mitochondrial networks. This toxicity is due mostly to PVD since a pvdF mutant which does not produce PVD is much less toxic to macrophages. Likewise, the anti-pyoverdine molecules 5FC and 5 FUR decrease PVD production and toxicity. To eliminate the contribution of toxins, such as exotoxin A or PrpL protease, the supernatants have been filtered to eliminate large proteins, with a similar effect. They extended and strenghtened their study by establishing a clear correlation between PVD production and cellular toxicity in 19 clinical isolates. The work is very well realized and the data are strong.
Comments:
- The contribution of pyochelin (PCH) to cellular toxicity has not been analyzed. PCH is known to generate reactive oxygen species and to be toxic to animal cells (see the articles of Britigan et al.). It is a pity that a double PCH PVD negative mutant has not been not used. The authors mention in lines 143-145 that other factors than PVD could be at work.
- Did the authors test for the production of pyocyanin in the serum free medium. I suppose that it must be negligible since none can be detected in the low iron CAA medium, just to confirm, since it is also toxic.
- The authors should mention in the Introduction that studies have been done on amoebas as well, as surrogates for macrophages.
Round 2
Reviewer 1 Report
The manuscript was edited and improved. The remaining minor suggestions are listed below.
- The details of treating cultures with inhibitors was not described in the methods.
- In the rebuttal, the authors provided a clear rational for using gallium, which, in my opinion, would be useful for the readers. I also think that the point of pyoverdine triggering a cascade of events increasing the overall virulence, which the authors provided in their responses, would benefit the discussion.
Author Response
Reviewer 1
Comments and Suggestions for Authors
The manuscript was edited and improved. The remaining minor suggestions are listed below.
- The details of treating cultures with inhibitors was not described in the methods.
We apologize for the oversight. These details are now included in section 4.1 of Materials and Methods (lines 338 - 339).
- In the rebuttal, the authors provided a clear rational for using gallium, which, in my opinion, would be useful for the readers. I also think that the point of pyoverdine triggering a cascade of events increasing the overall virulence, which the authors provided in their responses, would benefit the discussion.
We appreciate the suggestion. We have included our rational for using pyoverdine-gallium in lines 183 – 184 of the results section and further elaborated pyoverdine’s role in increasing bacterial virulence in lines 305 – 307 of the discussion section.